# *Rickettsia felis* and Other *Rickettsia* Species in Chigger Mites Collected from Wild Rodents in North Carolina, USA

**DOI:** 10.3390/microorganisms10071342

**Published:** 2022-07-02

**Authors:** Loganathan Ponnusamy, Reuben Garshong, Bryan S. McLean, Gideon Wasserberg, Lance A. Durden, Dac Crossley, Charles S. Apperson, R. Michael Roe

**Affiliations:** 1Department of Entomology and Plant Pathology, Comparative Medicine Institute, North Carolina State University, Raleigh, NC 27695, USA; charles_apperson@ncsu.edu (C.S.A.); michael_roe@ncsu.edu (R.M.R.); 2Department of Biology, University of North Carolina at Greensboro, Greensboro, NC 27402, USA; rngarsho@uncg.edu (R.G.); b_mclean@uncg.edu (B.S.M.); g_wasser@uncg.edu (G.W.); 3Department of Biology, Georgia Southern University, 4324 Old Register Road Statesboro, Statesboro, GA 30458, USA; ldurden@georgiasouthern.edu; 4Georgia Museum of Natural History, Natural History Building, University of Georgia, Athens, GA 30602, USA; crossley@att.net

**Keywords:** *Rickettsia*, chigger mites, North Carolina

## Abstract

Chiggers are vectors of rickettsial pathogenic bacteria, *Orientia* spp., that cause the human disease, scrub typhus, in the Asian–Pacific area and northern Australia (known as the Tsutsugamushi Triangle). More recently, reports of scrub typhus in Africa, southern Chile, and the Middle East have reshaped our understanding of the epidemiology of this disease, indicating it has a broad geographical distribution. Despite the growing number of studies and discoveries of chigger-borne human disease outside of the Tsutsugamushi Triangle, rickettsial pathogens in chigger mites in the US are still undetermined. The aim of our study was to investigate possible *Rickettsia* DNA in chiggers collected from rodents in North Carolina, USA. Of 46 chiggers tested, 47.8% tested positive for amplicons of the 23S-5S gene, 36.9% tested positive for 17 kDa, and 15.2% tested positive for *gltA*. Nucleotide sequence analyses of the *Rickettsia*-specific 23S-5S intergenic spacer (IGS), 17 kDa, and *gltA* gene fragments indicated that the amplicons from these chiggers were closely related to those in *R. felis*, *R. conorii*, *R. typhi*, and unidentified *Rickettsia* species. In this study, we provide the first evidence of *Rickettsia* infection in chiggers collected from rodents within the continental USA. In North Carolina, a US state with the highest annual cases of spotted fever rickettsioses, these results suggest chigger bites could pose a risk to public health, warranting further study.

## 1. Introduction

Chigger mites in the genera *Leptotrombidium* and *Herpetacarus* are known biological vectors of the causative agent of scrub typhus, *Orientia tsutsugamushi*, an obligate intracellular bacterium closely related to the genus *Rickettsia* (Rickettsiales) [1,2]. The disease is a significant source of morbidity and mortality, with an estimated one billion people at risk, and approximately one million cases reported each year [3]. Until recently, scrub typhus exclusively occurred within the so-called ‘Tsutsugamushi Triangle’, ranging from Pakistan in the west, to far-eastern Russia in the east, and to northern Australia in the south. However, scrub typhus was found recently in the Middle East, southern Chile, and Africa [4,5]. Scrub typhus was also thought to be caused by only one species of *Orientia* (*O. tsutsugamushi*); however, a new species, *Candidatus* Orientia chiloensis, was found in South America, where the first case of scrub typhus was confirmed on Chiloé Island in southern Chile [6,7]. In addition to the scrub typhus pathogen, several other rickettsiae known to infect humans, namely *Rickettsia japonica*, *R. akari*, *R. felis*, *R. conorii*, and *R. typhi*, were detected in chiggers removed from wild rodents in Korea [8]. In China, trombiculid mites collected from wild rodents were found to harbor several other pathogens, including *Orientia tsutsugamushi*, *Bartonella* spp., and *Rickettsia* spp. [9]. Recently, *Candidatus* Rickettsia colombianensi was also detected in chiggers collected from rodents in Brazil [10]. Despite the public health importance and increasing discovery of chigger-borne human disease outside of the Tsutsugamushi Triangle, the role of chiggers in human infection with rickettsial pathogens in the US is still unknown. Here, we report the identification of *Rickettsia felis* and other *Rickettsia* spp. in chiggers removed from rodents trapped in woodland and field habitats of public lands in North Carolina (NC), USA.

## 2. Materials and Methods

### 2.1. Site Description

Eight field sites were sampled, three state parks, three game lands, one community recreational green space (i.e., Clawson-Burnley Park, Boone, NC, USA) and one national forest (Chattahoochee National Forest, in the extreme northeastern corner of Rabun County, GA, USA) (Figure 1). All sites are within the greater southern Appalachian Mountains ecoregion, except Lake Norman State Park in North Carolina, USA. The general vegetation in all the sites was mixed woodland or hardwood forest containing tree species such as: oak, hickory, maple, pine, chestnut, birch, and beech. Some open meadows within the game lands and national forest were sampled, and these were characterized by ferns, herbs, grasses, and young regenerative trees. The forest of Clawson-Burnley Park (CBP) surrounds a mowed athletics field. At the Lake Norman State Park (LNSP), the collection site, was a meadow habitat at the forest edge; this was where *Sigmodon hispidus* were collected.

In every site, one or more of the following recreational activities may occur: hiking, kayaking, canoeing, fishing, rock climbing, hunting, birding, biking, camping, and picnicking. Hunting or extraction of other natural resources is only permissible, to variable extents, in the game lands, state forest (not the state parks), and national forest. Picnic-friendly microsites were established in designated places at some sites, either in the woods or open forest spaces, where picnic tables, shelters, and barbecue stands were constructed.

### 2.2. Rodent Trapping and Chigger Collection

Small mammals were live-trapped (except for one transect at CNF where snap traps were used) from June to September of 2020. Each trapping session consisted of three consecutive nights. Transect lines were established to cover different microhabitats within both forest and meadow. Live-trapping of small mammals primarily involved aluminum Sherman live animal traps (HB Sherman Traps Inc., Tallahassee, FL, USA), Longworth traps (Anglian Lepidopterist Supplies, Norfolk, England); pitfalls were also used at some sites. The use of different trap types increases the overall species richness by improving the chances of trapping species that may be easily caught by one trap type but not the other, and increases the overall capture efficiency [11]. Traps were baited with a mixture of peanut butter (Great Value Creamy Peanut Butter, Walmart Stores Inc., Bentonville, AR, USA) and oats (Great Value Old Fashioned Oats, Walmart Stores Inc., Bentonville, AR, USA). The traps were set at sunset and inspected at sunrise every day of the trapping period, closed after every morning inspection, and reopened again at sunset. Trapped small mammals were either anesthetized using the isoflurane vapor method before ectoparasite screening, or screened under manual restraint without anesthesia. Captured individuals were thoroughly inspected for the presence of chiggers and other ectoparasites by systematically examining the pelage, paying particular attention to the ears, feet, abdomen, and genitals, where blood-feeding ectoparasites are not easily dislodged [12]. Embedded chiggers appeared in clusters as red spots, and were removed using sterile pointed tweezers from their point of contact with the animal’s skin, or dislodged from the skin by vigorously brushing the pelage with a louse comb. All chiggers collected were attached to the animal skin. As part of ongoing projects, we also collected fleas, lice, and ticks using the same methods. All collected ectoparasites (including chiggers) were immediately transferred into 1.5-mL Eppendorf tubes or glass vials containing 95% absolute ethanol (Fisher Scientific, Hampton, NH, USA). One vial of ectoparasites was collected per individual small mammal, and these were stored at −20 °C in the laboratory for later identification.

Standard morphometric measurements were taken for each individual small mammal captured (i.e., body length and weight). Animals were also sexed (based on anogenital distance, which is wider in males, or visible evidence of descended testes or enlarged mammae), examined externally for reproductive status, and aged based on pelage and external traits, such as weight. Each animal was identified using the Peterson Field Guide to Mammals of North America, North of Mexico [13]. For uncertain identifications, DNA barcoding was used as a confirmatory test. To provide tissue for DNA barcoding analyses, we either used a 2-mm ear punch kit (Miltex Instruments, York, PA, USA) or subsampled liver tissue from animals collected lethally as part of other ongoing projects. Each tissue type was preserved in 1.5 mL vials containing 95% ethanol. Recaptures within each trapping session (identified by their previous ear punch, weight, transect line, and/or a mark made on the abdomen with a permanent marker) were re-examined for ectoparasites, reweighed, and released at the capture site. Small mammal field handling techniques followed the American Society of Mammalogists guidelines [14] and Institutional Animal Care and Use Committee (IACUC) protocols in place at the University of North Carolina at Greensboro (IACUC 18-002 and 20-008 for Wasserberg and McLean labs, respectively). Permits to conduct trapping at our sites were granted by the North Carolina Wildlife Resources Commission.

### 2.3. Extraction of DNA from Chigger Samples

To prevent cross-sample contamination, we performed template DNA extraction and PCR setup in separate rooms with dedicated pipette sets, used and certified DNase- and RNase-free filter barrier tips to avoid aerosol contamination. Total DNA was extracted separately from 46 individual chiggers collected from a portion of the *P. leucopus* and *S. hispidus* captured at the Lake Norman State Park and South Mountains Game Lands sites. To remove surface contaminants, including microbes on the surface of a chigger, prior to DNA extraction, each individual chigger was surface sterilized using methods described by Ponnusamy et al. [15]. The DNA of the chigger samples was extracted using the QIAGEN DNeasy Blood & Tissue Kit, based on the manufacturers’ instructions (QIAGEN, Valencia, CA, USA). Briefly, after surface sterilization, each chigger was transferred into a separate sterile 2-mL screw-capped microcentrifuge tube containing ~10 sterilized 3-mm glass beads (Cat. 11-312A, Fisher Scientific), with ATL lysis buffer and Proteinase K solution. Next, samples were homogenized using a FastPrep FP120 cell homogenizer (Thermo Electron Corporation, Waltham, MA, USA), and then incubated at 65 °C for 1 h. AL buffer was added to each tube, and the samples were mixed by pulse vortexing, and then incubated at 56 °C for 1 h. DNA was purified according to the manufacturers’ instructions. DNA was eluted using 30 µL of nuclease-free water and stored at −20 °C until further processing.

### 2.4. Identification of Chigger Samples

Molecular identification of chigger samples was conducted by PCR amplification of the mitochondrial cytochrome c oxidase subunit I gene (COI), using previously described primers and amplification protocols [16]. Twenty percent of amplicons were Sanger sequenced at Eton Bioscience, Inc. (Research Triangle Park, NC, USA). Morphologically based identifications were conducted using expert knowledge and published taxonomic keys [17,18,19]. A portion of the chiggers collected from South Mountains Game Lands were submitted as voucher specimens to the UNCG Parasite Collection. Similarly, Lake Norman State Park chigger subsamples were submitted to the Georgia Museum of Natural History.

### 2.5. Amplification of Rickettsia spp. from Chigger DNA

We screened the subset of chigger samples by amplifying the 23S-5S intergenic spacer (IGS) [20,21], and the 17 kDa [22] and *gltA* genes [23,24], using the *Rickettsia*-specific primers provided in Table 1. The PCR reaction mixture comprised of 10 μL of 2X AmpliTaq Gold PCR Master Mix (Applied Biosystems, Grand Island, NY, USA) and 1 μL each of forward and reverse primers (10 μM), with 1 μL of genomic DNA in the primary reactions. One μL of the amplicon from the primary PCR was used as a template in subsequent nested PCR assays, and sterile deionized water was added to achieve a final volume of 20 μL. Negative controls (reaction mixture without template DNA) were used in every set of PCR assays. Previously published protocols for *Rickettsia*-specific nested PCR primers and assay conditions were used to amplify the 23S-5S [20], and 17 kDa, and *gltA* [25] regions. For 23S-5S IGS PCR amplification, the first round of amplification conditions used were 95 °C for 10 min, 35 cycles of 94 °C for 30 s, 60 °C for 30 s, and 65 °C for 90 s, and a last cycle of 65 °C for 7 min. The second nested amplification conditions used were 95 °C for 10 min, 30 cycles of 94 °C for 30 s, 57 °C for 30 s, and 72 °C for 90 s, and a last cycle of 72 °C for 10 min. For the 17 kDa protein gene, conditions for the first round of amplification were 10 min at 95 °C, followed by 35 cycles of 95 °C for 30 s, 47 °C for 30 s, and 72 °C for 60 s, and a last cycle of 72 °C for 10 min. The nested PCR conditions were similar to those of the first round of PCR amplification, except that the annealing temperature was increased to 50 °C, and the cycles were repeated 30 times. The PCR conditions used for *gltA* were at 95 °C for 10 min, followed by 30 cycles of 94 °C for 30 s, 57 °C for 30 s, 72 °C for 60 s, and a last cycle of 72 °C for 5 min. The nested reaction included 95 °C for 10 min, 30 cycles of 94 °C for 30 s, 55 °C for 30 s, and 72 °C for 60 s, and a final cycle of 72 °C for 5 min. Two independent PCR assays were performed for each *Rickettsia* sp. for all three targets. The amplicons were subjected to electrophoresis on ethidium bromide-stained 1.2% agarose–Tris-acetate-EDTA gels. The amplicons were purified and subjected to Sanger sequencing using the forward primer from the nested reactions at Eton Bioscience, Inc. (Research Triangle Park, NC, USA).

### 2.6. Phylogenetic Analyses of Rickettsia Sequences

BLASTn and nucleotide sequence match analyses were used to compare partial nucleotide sequences of the 23S-5S IGS, 17 kDa, and *gltA* amplicons to those in the GenBank database. Sequences were aligned with the multiple alignment CLUSTALX software package [26]. Evolutionary distances between sequences of the 23S-5S, 17 kDa, and *gltA* amplicons and known *Rickettsia* species from the NCBI database were calculated (Kimura two-parameter model) [27], and phylogenetic trees were constructed separately for each region using the neighbor-joining (NJ) and maximum likelihood (ML) methods [28,29] from the MEGA X software package [30]. Bootstrap analyses, consisting of 1000 iterations, were performed to evaluate the robustness of tree topologies. All three *Rickettsia*-specific PCR targets were successfully amplified from chigger DNA samples C6, 12, 22, and 36, and single PCR targets were amplified from several additional samples. To further refine the taxonomic positions of these rickettsiae, sequences of the 23S-5S IGS, 17 kDa, and gltA genes were retrieved from the NCBI whole genome assemblies, and combined phylogenetic analyses were performed with concatenated sequences using MEGA X software. Evolutionary distances were calculated using the Kimura two-parameter model, and phylogenetic trees were inferred with NJ and ML analyses using MEGA X software, as described above.

## 3. Results

### 3.1. Chigger Infestation of Rodents

Chigger specimens were collected from two species of rodents at the two sites surveyed. The subsample of chiggers analyzed in the present study were removed from white-footed mice (*Peromyscus leucopus*), trapped in the South Mountains Game Lands in a typical hardwood forest habitat, and a cotton rat (*Sigmodon hispidus*) from Lake Norman State Park in a grassy field habitat near the edge of the forest. A total of 109 larval chigger mites of two different species were collected. The sequences submitted for BLASTn analyses showed 85 to 91% identity with homologous sequences from Trombiculidae species for the COI gene (GenBank accession number MG314759). Based on their morphology, chiggers collected from *P. leucopus* at South Mountains Game Lands were identified as *Leptotrombidium peromysci*, and those collected from *S. hispidus* at Lake Norman were in the genus *Eutrombicula* (Table 2).

### 3.2. Rickettsia Infection Patterns

Amplification details of the three gene/23S-5S IGS targets for detection of *Rickettsia* spp. in chiggers are listed in Table 1. Twenty-two of the 46 chiggers we tested (47.8%) were positive for *Rickettsia* using the 23S-5S gene. Of those, we sequenced all of the 23S-5S amplicons and compared them by BLASTn to similar sequences in GenBank. Of the 22 amplicons, five sequences (C6, 12, 22, 47, 48) were 99.5–100% homologous with *R. felis* (NC007109). One amplicon (C8) had a 98.6% identity with *R. conorii* (NC003103), two amplicons (C7 and C33) had an 86.9% identity with *R. amblyommatis* (CP015012), and one amplicon (C36) had a 93.3% identity with *R. typhi* (NC017066). The remaining thirteen amplicons had ambiguous sequences (likely indicating more than one species of *Rickettsia* present).

The NJ phylogenetic tree based on 23S-5S IGS from *Rickettsia* amplicon sequences revealed that the *Rickettsia* detected in the chiggers were closely related to *R. felis*, *R. conorii*, *R. typhi*, and unidentified *Rickettsia* species (Figure 2). Similar results were obtained using the ML method (Appendix A).

From the twelve 17 kDa amplicons, five had a 99.1 to 100% identity with *R. felis* (NC007109), and one had a 99.3% identity with *R. amblyommatis* (CP015012). Six of the amplicons had ambiguous sequences. NJ-based phylogenetic analyses of the 17 kDa gene sequences showed that *Rickettsia* spp. in chiggers were closely related to *R. felis* and *R. amblyommatis* (Appendix A). These same relationships were also reflected in the ML tree (Appendix A). Of the seven *gltA* amplicons, five showed 99 to 99.8% similarity with published sequences for *R. felis*, one sequence showed 99.8% similarity to *R. amblyommatis*, and one amplicon had ambiguous sequences. NJ-based phylogenetic analyses based on five *gltA* gene sequences (C6, C12, C22, C36, and C48) were closely related to *R. felis,* one sequence (C7) was closely related to *R. amblyommatis* (Appendix A); the same relationships were also reflected in the ML tree (Appendix A). Only four samples (C6, 12, 22, and 36) produced amplicons for all three PCR targets. The phylogenetic analysis with concatenated sequences, 23S-5S IGS, 17 kDa, and *gltA*, showed that chigger amplicon sequences C12 and 22 were clustered in the same clade with *R. felis*, C6 was sister to the *R. felis* clade, and C36 to the *R. typhi* clade (Figure 3). Similar results were found using the ML method (Appendix A).

## 4. Discussion

In this study, we provide the first evidence of *Rickettsia* infection in chiggers collected from rodents within North America, a significant extension of recent work finding rickettsial DNA in trombiculid mites from South America and Asia [10,31,32]. Nucleotide sequence analyses of *Rickettsia*-specific 23S-5S IGS, 17 kDa, and *gltA* gene fragments indicated the amplicons from these chiggers were closely related to those in *R. felis*, *R. conorii*, *R. typhi*, and unidentified *Rickettsia* species. The most frequently detected rickettsial species in our study, *Rickettsia felis*, is globally distributed and was found in fleas, mosquitoes, ticks, and chigger mites [31]. There is increasing evidence that chiggers could be vectors of transitional group rickettsiae, because of their discovery in a variety of field-collected chiggers [10,32]. Several other rickettsiae known to infect humans, namely *R. japonica*, *R. akari*, *R. felis*, *R. conorii*, *R. typhi*, and other *Rickettsia* spp. closely related to TwKM02 (*R. australis* and Cf15) were detected by molecular methods in chiggers collected from wild rodents in Korea [8]. Interestingly, chiggers in the US feed on the same rodent species that serve as blood meal hosts for ticks known to carry rickettsial species that cause spotted fever-type diseases, further raising the question of whether some infections attributed to ticks could in fact be derived from chiggers.

Historically, chigger mites were collected from rodents captured in the region of the ‘Tsutsugamushi Triangle’, in areas of Korea, Japan, throughout southern Asia, the Asian–Pacific region, and northern Australia [33,34]. Scrub typhus was thought to be exclusive to the so-called ‘Tsutsugamushi Triangle’, ranging from Pakistan in the west, far-eastern Russia in the east, to northern Australia in the south. However, recent scrub typhus reports in the Middle East, southern Chile, and Africa have reshaped this line of thinking, suggesting a wider geographical distribution of the disease [3,4,6,35,36,37,38]. The vectors of scrub typhus caused by *Orientia tsutsugamushi* (closely related to the genus *Rickettsia*, i.e., Rickettsiales: *Rickettsiaceae*) [1,2] are chiggers in the genus *Leptotrombidium* in the Tsutsugamushi Triangle [33,34], and in the genus *Herpetacarus* in Chile [2,6]. Both *Leptotrombidium* and *Eutrombicula* collected from rodents in our study in the US (known to bite humans, causing skin inflammation and hypersensitivity lesions [39,40]) were found infected with *Rickettsia* species. Additional epidemiologic studies are needed to examine whether there is a link between these rickettsiae-infected mites on wild rodents and human disease. An examination of *Orientia* species in US chiggers is also needed.

Based on morphological identification, two trombiculid genera, *Leptotrombidium* and *Eutrombicula,* were found on rodents in our study. However, our COI sequence data from these chiggers had no significant match with these genera. The BLAST search only matched with Trombiculidae. The lower sequence identity with the latter could be due to the lack of sequence information for US chiggers in GenBank. Presently, molecular data on chiggers in GenBank are mostly from southeast Asian and South American countries. Clearly, there are concerns about the present state of systematics of US chiggers that will require a significant level of future research effort to resolve (and which is outside the scope of this paper). Our study is the first to identify the genus *Rickettsia* in US chigger mites from rodents. Future studies to determine rickettsial infection in other ectoparasites on the same host, in hosts tissues, and in free-living chiggers, in order to understand possible vertical, transovarial transmission of rickettsiae would be informative.

## Figures and Tables

**Figure 1 microorganisms-10-01342-f001:**
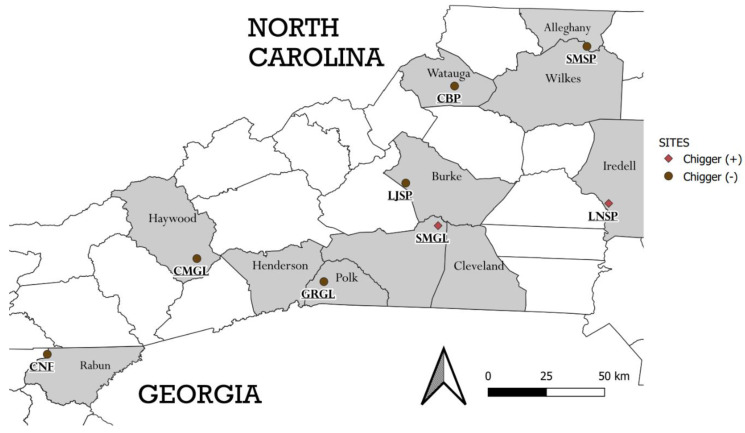
Map of areas where small mammals were sampled for ectoparasites. Shaded counties depict areas where rodents were trapped. Brown circles indicate areas where no chiggers were found on rodents, and red diamonds indicate sites where chiggers were found on some rodents. Some parks span beyond one county (indicated below by asterisks) and trapping extended into the nearby counties. * SMSP—Stone Mountain State Park (Alleghany and Wilkes counties), CBP—Clawson-Burnley Park, LJSP—Lake James State Park, LNSP—Lake Norman State Park, * SMGL—South Mountains Game Lands (Rutherford and Cleveland), CMGL—Cold Mountain Game Lands, * GRGL—Green River Game Lands (Polk and Henderson), CNF—Chattahoochee-Oconee National Forest.

**Figure 2 microorganisms-10-01342-f002:**
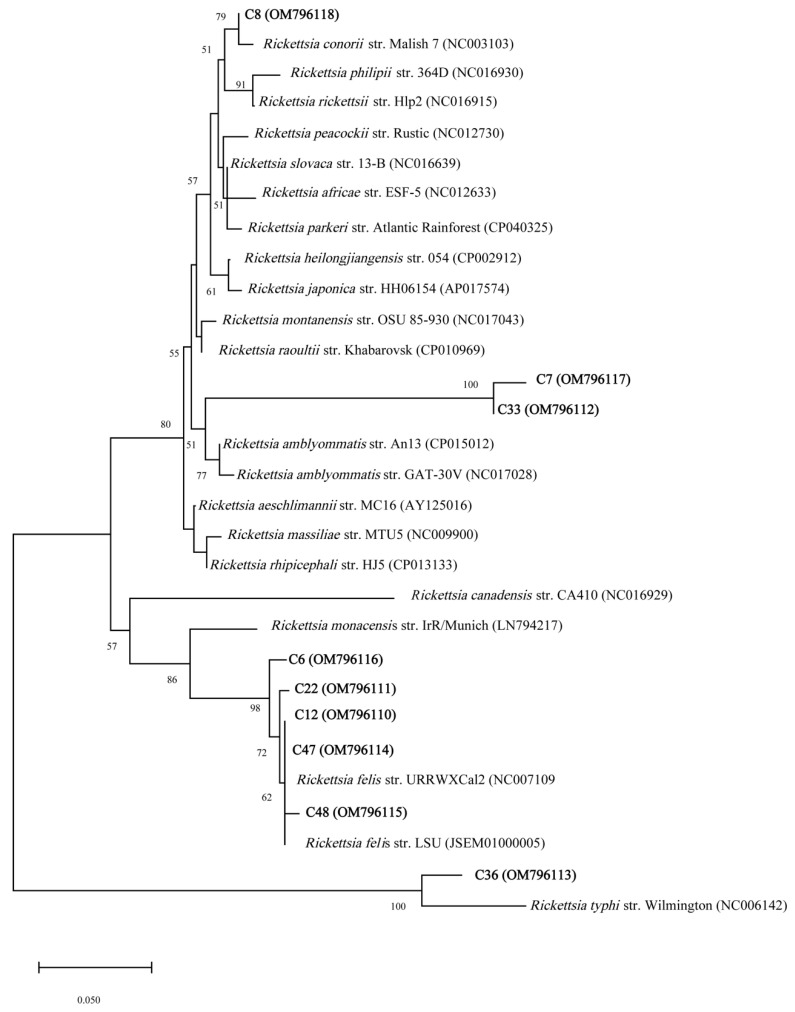
Phylogenetic relationships derived from partial sequences (~231 bp) of the *Rickettsia* 23S-5S IGS amplified from chigger DNA samples (highlighted in bold) and other related *Rickettsia* taxa, inferred using the NJ method. The GenBank accession numbers are given in the parentheses. Scale bars indicate the number of substitutions per nucleotide position.

**Figure 3 microorganisms-10-01342-f003:**
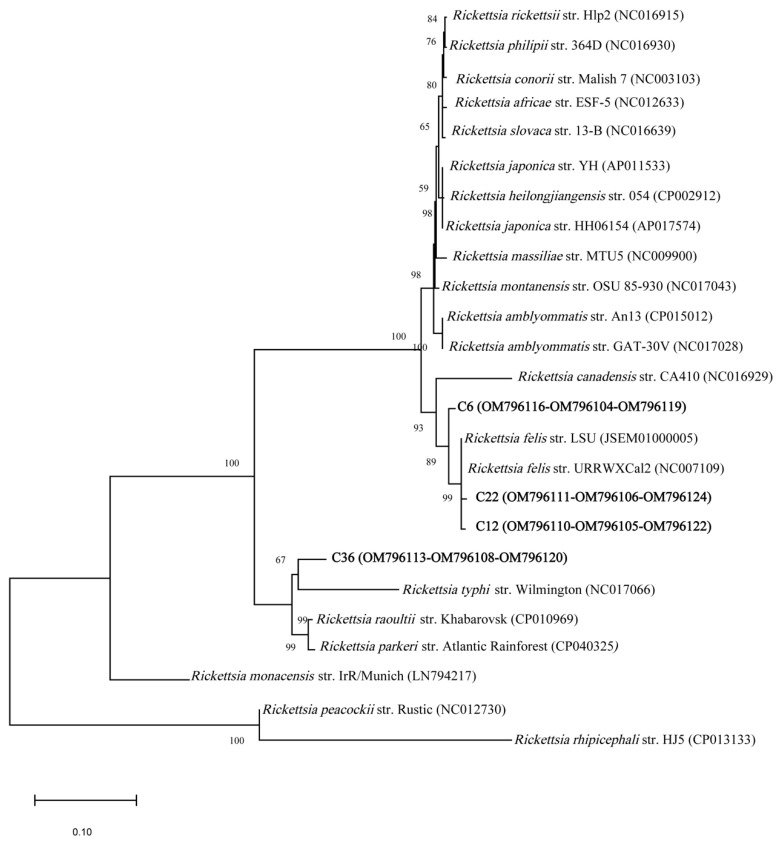
Phylogenetic tree of *Rickettsia* sequences from chiggers (bold letters) collected from rodents in North Carolina, USA, and reference sequences in GenBank (NCBI). The tree was inferred from the concatenated sequences (~733 bp) of three regions (23S-5S IGS, 17 kDa, and *gltA*) of the *Rickettsia* spp. genome by the neighbor-joining (NJ) method. The GenBank accession numbers are given in parentheses.

**Table 1 microorganisms-10-01342-t001:** Nucleotide sequences of primers used in nested PCR.

Organism (References)	Primers	Sequence (5′ to 3′)	Amplification	Target Gene (Nested Amplification Product Size)
*Rickettsia* [20,21]	RCK/23-5-F	GATAGGTCRGRTGTGGAAGCAC	Primary	23S-5S target (~350 bp)
	RCK/23-5-R	TCGGGAYGGGATCGTGTGTTTC	Primary	
	RCK/23-5N1F	TGTGGAAG CACAGTAATGTGTG	Nested	
	RCK/23-5N1R	TCGTGTGTTTCACTCA TGCT	Nested	
*Rickettsia* [22]	17kD1_F	GCTCTTGCAACTTCTATGTT	Primary	17 kDa protein gene (232 bp)
	17kD2_R	CATTGTTCGTCAGGTTGGCG	Primary	
	17kN1_F	CATTACTTGGTTCTCAATTCGGT	Nested	
	17kN2_R	GTTTTATTAGTGGTTACGTAA	Nested	
*Rickettsia* [23,24]	RpCS.877p	GGGGGCCTGCTCACGGCGG	Primary	*gltA* (338 bp)
	RpCS1258n	ATTGCAAAAAGTACAGTGAACA	Primary	
	RpCS896p	GGCTAATGAAGCAGTGATAA	Nested	
	RpCS1233n	GCGACGGTATACCCATAGC	Nested	

**Table 2 microorganisms-10-01342-t002:** Molecular detection of genus *Rickettsia* spp. in chigger mites.

Location *	Chigger-Infested Rodent Species	Species of Chigger	Number of Chiggers Screened	Number of Chiggers Positive for the Gene Fragment		
				23S-5S ^#^	17 kDa ^#^	*gltA* ^#^
*LNSP*	*Sigmodon hispidus*	*Eutrombicula* spp.	17 (1)	9	4	3
*SMGL*	*Peromyscus leucopus*	*Leptotrombidium peromysci*	29 (4)	13	8	4
Total			46	22	12	7

* LNSP—Lake Norman State Park, SMGL—South Mountains Game Lands. ^#^ Samples were considered positive for genus *Rickettsia* spp. only when DNA of the expected size was amplified for a specific PCR target in both replicate assays. Numbers in parentheses indicate the number of individual rodent species from which chiggers were collected.

## Data Availability

The gene data presented in this study are openly available in the NCBI GenBank under OM796104-OM796124.

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
