# Peer review of "Rickettsia felis and Other Rickettsia Species in Chigger Mites Collected from Wild Rodents in North Carolina, USA"

_microorganisms, 2022, doi:10.3390/microorganisms10071342_

Round 1

Reviewer 1 Report

This is fairly straightforward but interesting manuscript reporting the first detection of Rickettsia spp. in trombiculid mites in the continental US. It is timely, following on the heels of several similar reports from other parts of the world demonstrating that the prevalence of Rickettsia in chiggers has been overlooked due to a narrow focus on Orientia spp. Overall, the study has been conducted appropriately but there are some important methodological shortcomings, as well as some deficiencies in the presentation of results and discussion.

Important points: 

1.  The site selection and rodent trapping methods sections are quite lengthy and include a lot of detail that would be appropriate for an ecological study but not this particular report. No doubt the authors plan to publish more data from the rodent trapping fieldwork, but in this manuscript, the results do not include any form of ecological analysis or risk mapping exercise, and minimal information is provided on the rodents collected. Please just include the essential points in the methods regarding site selection and trapping rather than details of transect design.

2. The chigger barcoding doesn't really add any information, which is not surprising, since very little molecular data for chiggers is publicly available for comparison. The generated sequences are not presented or discussed, and appear to be too distant from existing sequences to be of any use for identification purposes. These data should be omitted unless the authors can demonstrate, using the sequences they have generated and prior sequences, that they tell us something useful. 

3. Related to the point above, the limitations of the the study should be openly discussed with respect to chigger ID. As the same individual chiggers were not identified and then subjected to DNA extraction (which is possible - see for instance https://doi.org/10.1371/journal.pone.0193163), then the identify of the infected chiggers is provisional. Many studies worldwide have reported co-infection of hosts with multiple chigger species. The coi barcodes for the chiggers could potentially be used to verify that infected chiggers were the expected species (through clustering of the coi barcodes and association with infection at an individual level). If this was actually done, please be explicit.

4. The length of the alignments (nucleotides) used for each tree should be included in the tree legends.

5. The authors frequently refer to the chigger-derived Rickettsia sequences being close to unidentified Rickettsia spp., although all sequences included in the trees are from named Rickettsia spp. Were better matches to unidentified species sometimes seen in BLAST? The authors may have decided not to include certain sequences in their trees (e.g., data in GenBank lacking an associated publication) but if so, please discuss this.

6. The discussion is a bit perfunctory. Please expand on how the importance of chigger-derived Rickettsia could be determined and the biology probed further. For instance, in the recent Taiwanese study (DOI: 10.1111/mve.12560), the authors included data on whether the Rickettsia spp. in chiggers were also found in other ectoparasites on the same host and/or in host tissues. The authors of the current manuscript would seem to have the specimens to do this themselves in future - please explain. Also, an important study from Japan (https://doi.org/10.1111/1348-0421.12745) should be cited in the Discussion, as it shows that Rickettsia can be found in unfed chiggers, indicating vertical transmission can occur.   

7. In the discussion, the impression is given that most Rickettsia detected in chiggers are from the Spotted Fever Group. However, many (perhaps most if all published data are considered in aggregate) are closer to the Transitional Group (R. felis, R. australis etc.). Please correct this.

Minor points:

1. In the introduction, the authors state that Leptotrombidium spp. are the sole biological vectors of scrub typhus. It is known that the vector in Chile is not Leptotrombidium spp., but Herpetacarus spp. (https://doi.org/10.1093/cid/ciab748)

2. There are very few studies of Rickettsia in chiggers, so the authors can afford to be more comprehensive in the Introduction. There is a prior study from China not cited (https://doi.org/10.1093/jme/tjw234), as well as the one from Japan mentioned above. 

3. Figure legends should be separated from the main text by a blank line; also for tables.

4. In Fig. 1, the significance of the grey shading is unclear. Sampling does not seem to have taken place in every shaded county? 

5. The species of rodents trapped are reported in the methods, but this seems unnecessary - it should be part of the results.

6. In table 2, part of the second row is in bold type, but it is unclear why. The no. of rodents involved is also unclear - it seems to be one cotton rat but several mice? Please clarify the no. of mice with infected chiggers.

7. Be consistent with gene terminology. Italicise gltA in all cases. The 23S-5S IGS is not always referred to as such (and intergenic spacer needs to be defined on first use for people not familiar with the abbreviation).

8. The sentence "NJ based phylogenetic analyses based on gltA gene sequences (Supplementary Figure S4) and tree (Supplementary Figure S5)” seems incomplete (lines 248 -249).

9. The list of supplementary materials after the Discussion section is incomplete.

10. Line 42. Candidatus is spelled incorrectly. 

Reviewer 2 Report

I carefully read the manuscript entitled “Rickettsia felis and other Rickettsia Species in Chigger Mites collected from wild rodents in North Carolina, USA” 

The MS is well written and structured. The study and the obtained results are interesting but the discussions are limited. The authors don’t describe the limitations of the study. For instance, the ear biopsy of the rodents or tissue sample were available but the authors didn’t test them to obtain more epidemiological data by comparations. However, these results are still important in the field and I support the publication of the MS, after the authors discuss more deeply their findings. 

Author Response

This manuscript is a resubmission of an earlier submission. The following is a list of the peer review reports and author responses from that submission.

Round 1

Reviewer 1 Report

The reviewer’s comments and suggestions for Authors are as follows:

Materials and Methods

1. Rodent trapping and collection of chigger mites

Authors need clearly indicate the number of rodents trapped form 2 collection sites for scientific reason.

2. Amplification of Rickettsia spp. from chigger DNA

  1. In order to let readers directly understand that the author needs to add a description of the brand of the PCR reagent kit used and PCR condition (Ex. Annealing temp., PCR cycles et al.) in the content of the article.
  2. In addition to the format of Table 1, it needs to be rearranged so that each individual primer sequence can be clearly displayed on the same line.

3. Phylogenetic analyses of Rickettsia sequences

Usually two different phylogenetic computational methods are used to obtain the same result, which can be proved the strong and powerful reliability for constructing phylogenetic trees of Rickettsia specific gene sequence. However, the author construct phylogenetic trees of Rickettsia spp. by only neighbor-joining method from the MEGA X software. Thus, reviewer suggests that author should to do another phylogenetic computational method (such as maximum-likelihood or maximum-parsimony method) to construct phylogenetic trees to make sure the Phylogenetic analyses of Rickettsia.

Results

1. Chigger infestation of local rodents

  1. Base on the author’s molecular identification result of chiggers, “the 121 sequences submitted for BLASTn analyses showed 85 to 91% identity with homologous 122 sequences from Trombiculidae species for the COI gene (GenBank accession number MG314759)”. However, that just indicate the COI gene homologous of Trombiculidae by family level, not species level. Therefore, author should to compare the COI gene sequence of Eutrombicula and Leptotrombidium spp. mite with GenBank accession number MK015032 and HQ324987, respectively. Otherwise results of this article can’t support the molecular evidence of species level for chigger mites.
  2. Table 2 needs to be rearranged and merge the datas of same sample source informations: such as location (SMGL)、Chigger infested rodent species (Peromyscus leuco-pus) and Species of chigger (Leptotrombidium peromysci).

2. Rickettsia infection patterns

  1. Base on author’ s description in Materials and Methods (Line 104-105), they construct phylogenetic trees of Rickettsia spp were made separately for each region (23S-5S, 17 kDa and gltA amplicons) only using the neighbor-joining method [23] from the MEGA X software. However, the text of the manuscript presents the “maximum-likelihood method” of phylogenetic trees (Figure 3) that make reviewer feel so confuse.
  2. The figure legends of Figure 2 need clearly to describe the ” phylogenetic computational methods” (ex: maximum-likelihood method et al.)
  3. In order to enhance the data reliability and present geographic affiliation, reviewer strongly suggests author have to use (or add) some felis & R. amblyommatis related sequences of USA isolates from NCBI, and reconstruct the phylogenetic tree for each region (23S-5S, 17 kDa and gltA amplicons) including Figure2 & 3.

Reviewer 2 Report

This manuscript describes the detection of Rickettsia in chiggers collected in North Carolina (USA).  The work presented is important from a public health perspective, as chiggers are apt to bite humans.  Moreover, the recent reports of Orientia in South America warrant studies to detail the presence of potential human pathogens in chigger populations in the Western hemisphere.

Comments/suggestions:

Introduction:

L45:  Full genus/species should be written out for the first mention of an organism.

Materials and Methods:

Section 2.1: 

-- A detailed description of the sites that were used for trapping would enhance the manuscript.  Were the sites frequent recreation areas, are they forested (controlled), are they mowed, are they regularly burned, etc.?

--  More details concerning the trapping procedure would enhance the manuscript.  How often were traps set from 2019-2020 - monthly, weekly?  How many traps per site were set?  How many sites were set for trapping within each location?  Were both sites trapped at the same time?

-- Were the animals anesthetized for chigger collection?  Or were they euthanized?  The details should be provided.

Section 2.2

-- The extraction method that was used should be mentioned (beyond just a citation)

-- L72:  Why was DNA extracted only from a portion of the chiggers?

Section 2.3

--L82-84:  Provide the appropriate references within the text.  

-- What mastermix/PCR kit was used?

-- Were any other genes (with a longer target sequence) attempted - i.e. sca4, ompA, ompB?

Table 1:

-- Revise for primer sequences to each be on a single line.

Results:

-- How many total chiggers were collected?

-- How many total rodents were collected?

-- How many rodents had chiggers removed?

-- Were other ectoparasites removed from the rodents?

-- Were the rodents sexed? aged?

-- Were any chiggers positive for all three genes?  Or was each gene fragment obtained from a single chigger?

Figure 1:

-- Figure legend should be placed below the figure.

-- Additional/more detailed maps of the actual trapping sites would be helpful to present a more comprehensive overview of the study area.

L151-152:  Italicize "R. felis".

L170-171:  Full genus/species should be written out for the first mention of an organism.